# SR-Reward: Taking The Path More Traveled

**Seyed Mahdi B. Azad**                                           *basiri@cs.uni-freiburg.de*
*Department of Computer Science*
*University of Freiburg*

**Zahra Padar**                                                  *padarz@informatik.uni-freiburg.de*
*Department of Computer Science*
*University of Freiburg*

**Gabriel Kalweit**                                              *kalweitg@cs.uni-freiburg.de*
*Department of Computer Science*
*University of Freiburg*

**Joschka Boedecker**                                            *jboedeck@informatik.uni-freiburg.de*
*Department of Computer Science*
*University of Freiburg*

**Reviewed on OpenReview:** *https://openreview.net/forum?id=bzk1sV1svm*

## Abstract

In this paper, we propose a novel method for learning reward functions directly from offline demonstrations. Unlike traditional inverse reinforcement learning (IRL), our approach decouples the reward function from the learner's policy, eliminating the adversarial interaction typically required between the two. This results in a more stable and efficient training process. Our reward module, *SR-Reward*, leverages successor representation (SR) to encode a state based on expected future states' visitation under the demonstration policy and transition dynamics. By utilizing the Bellman equation, SR-Reward can be learned concurrently with most reinforcement learning (RL) algorithms without altering the existing training pipeline. We also introduce a negative sampling strategy to mitigate overestimation errors by reducing rewards for out-of-distribution data, thereby enhancing robustness. This strategy introduces an inherent conservative bias into RL algorithms that employ the learned reward, encouraging them to stay close to the demonstrations where the consequences of the actions are better understood. We evaluate our method on D4RL as well as Maniskill Robot Manipulation environments, achieving competitive results compared to offline RL algorithms with access to true rewards and imitation learning (IL) techniques like behavioral cloning. Code available at: `https://github.com/Erfi/SR-Reward`

## 1 Introduction

Imitation learning (IL) from expert demonstrations is a widely used approach for tackling sequential decision-making tasks. Methods in this domain generally fall into two categories. The first focuses on directly learning a policy that mimics expert behavior, such as Behavioral Cloning (BC) (Pomerleau, 1991). The second, inverse reinforcement learning (IRL) (Ng & Russell, 2000), infers a reward function that explains an expert's behavior and simultaneously derives a policy from it. While both approaches have shown promise, they come with notable limitations. BC struggles with distribution shift, failing when encountering states not covered in the demonstrations (Ross et al., 2010). IRL, while more flexible, often inherits the instabilities of adversarial training (Goodfellow et al., 2014) and requires environment interaction to refine the learned reward function — an impractical requirement in domains where exploration is costly or unsafe, such as robotics and healthcare.

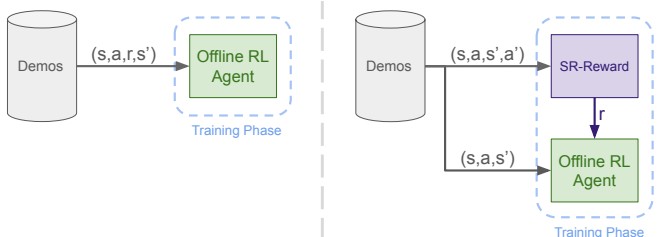

Figure 1: Standard offline RL training requires rewards to be present in the demonstrations (Left). SR-Reward enables the use of offline RL algorithms in settings where the demonstrations do not include rewards (Right). Adding the next action for training the SR-Reward requires only a small modification to the replay buffer.

Offline reinforcement learning (RL) provides an alternative by learning policies from fixed datasets without interacting with the environment (Lange et al., 2012). However, offline RL typically requires well-defined reward signals, which are often unavailable or difficult to engineer in real-world applications. Without a reward function, the agent has no clear objective and fails to learn a meaningful policy.

To address this challenge, we propose SR-Reward, a reward module that enables the use of offline RL agents in the absence of explicit reward functions. SR-Reward is based on the observation that frequently occurring features in expert demonstrations often capture essential aspects of expertise. By leveraging successor representations (SR)(Dayan, 1993) to estimate feature frequencies, SR-Reward assigns higher rewards to frequently observed state-action pairs. This provides a structured learning signal that helps agents align with expert strategies while retaining the flexibility to adapt to novel situations.

Successor representation provides a way to represent a state based on the frequency of its future visitation. In this work, we leverage SR as the primary mechanism to track how often state-action features appear in the demonstrations. Leveraging the SR structure allows SR-Reward to be learned via the Bellman equation which propagates information about future states and actions through temporal difference (TD) learning. Consequently, it can be integrated into existing training pipelines alongside other RL methods based on TD learning with minimal modifications. Figure 1 illustrates the role of SR-Reward within the training pipeline, highlighting how it integrates into the overall learning process. Unlike adversarial schemes popular with inverse reinforcement learning (IRL) methods (Ng & Russell, 2000; Abbeel & Ng, 2004; Ho & Ermon, 2016), our reward function is decoupled from the policy that is being learned. Decoupling the reward from the policy eliminates the instabilities associated with adversarial training and enables the use of a wide range of RL algorithms that were previously unusable due to the inaccessibility of the reward function (Fujimoto et al., 2019; Garg et al., 2023; Sikchi et al., 2023; Xu et al., 2023).

Hand-engineering a reward function may work for simple scenarios, but it becomes both challenging and error-prone in complex real-world tasks, such as robot manipulation (Singh et al., 2009; Wu et al., 2022). In contrast, demonstrating the desired behavior is generally more straightforward (Wu et al., 2023; Arunachalam et al., 2022; Rakita et al., 2017). It is in such scenarios, that SR-Reward's ability to learn a dense reward function from demonstrations can be valuable. Another avenue for using the demonstrations is to copy the expert's actions. Simple imitation learning methods like behavioral cloning (BC) directly mimic expert behavior without modeling how actions lead to future states. Unfortunately, this short-sighted objective is prone to distribution shift when encountering unseen states. SR-Reward enables the use of TD learning methods, which mitigate this issue by gaining a long-term view of the task via bootstrapping, making them more resilient to out-of-distribution scenarios.

We use function approximation to implement the SR mechanism in continuous state and action settings. Function approximation can lead to a significant overestimation of values for out-of-distribution data (Thrun & Schwartz, 1999). This is particularly problematic in our setting as the expert demonstrations by definition cover only a narrow subset of the overall space. We introduce a negative sampling strategy designed to counteract the overestimation error in our reward function for out-of-distribution states and actions. This is

accomplished by augmenting the Bellman loss for SR-Reward so that reward estimates for out-of-distribution states and actions decrease based on their distance from expert demonstrations. Incorporating negative sampling not only enhances the robustness of the reward function but also introduces a natural conservatism into the value functions and policies that rely on it. This approach encourages agents to remain near the demonstrated behaviors, where the outcomes of their actions are better understood.

In this work, we focus on the offline inverse reinforcement learning setting, where the agent neither has access to the reward function nor can query the expert for any feedback. Furthermore, the transition dynamics of the environment are unknown and the agent is provided with limited data in the form of expert demonstrations.

In summary our key contributions are:

1. SR-Reward: a reward function based on Successor Representation (SR), that can be learned solely offline using temporal difference (TD) learning simultaneously with other RL algorithms. We further describe our architecture as well as loss functions needed for training SR-Reward.

2. A negative sampling strategy to mitigate overestimation errors by reducing rewards for out-of-distribution data in the vicinity of the expert demonstrations, thereby increasing robustness by introducing a conservative bias into RL algorithms that employ the learned reward.

## 2 Background

We first introduce the notation and provide a more detailed review of concepts from successor representation and imitation learning.

### 2.1 Notation

We consider settings where the environment is represented by a Markov Decision Process (MDP) and is defined as a tuple $\mathcal{M} = (\mathcal{S}, \mathcal{A}, \mathcal{T}, r, \gamma, \mu_0)$. $\mathcal{S}$ and $\mathcal{A}$ represent the continuous state and continuous action spaces respectively. $\mathcal{T}(s'|s, a)$ represents the state transition dynamics, $r(s, a)$ represents the reward function, $\gamma \in (0, 1]$ is the discount factor and $\mu_0$ represents the starting state distribution. In the offline inverse reinforcement learning setting, we only have access to a limited set of expert demonstrations of the form $\mathcal{D} = \{(s_0, a_0, s_1, a_1, ...s_T)^i\}_{i=0}^{N}$. In this paper, we are focusing on a limited setting where neither the transition dynamics $\mathcal{T}(s'|s, a)$ nor the reward function $r(s, a)$ are known. The goal is to learn a reward function $r_\theta(s, a)$ from expert demonstrations such that its corresponding policy $\pi_\phi(a|s)$ performs similarly to that of the expert.

### 2.2 Successor Representations

Successor Representation (SR) was originally introduced as a method to generalize the value function across different rewards (Dayan, 1993). SR is defined as the cumulative discounted probability of visiting future states when following a specific policy, effectively representing the current state (and action) in terms of potential future states (and actions).

For any given pair of states $s, s'$ and actions $a, a'$, the SR is expressed as:

$$M(s, a, s', a') = \mathbb{E}\left[\sum_{t=0}^{\infty} \gamma^t \mathbb{I}(s_t = s', a_t = a')|s_0 = s, a_0 = a\right],$$

where the expectation is taken over the policy $\pi(a|s)$ and the environment's transition dynamics $\mathcal{T}(s'|s, a)$. Similar to the Q-function, SR can be estimated using the recursive Bellman equation:

$$M(s_t, a_t, s', a') = \mathbb{I}(s_t = s', a_t = a') + \gamma \mathbb{E}\left[M(s_{t+1}, a_{t+1}, s', a')\right].$$

This recursive formulation is particularly useful when learning SR alongside other temporal difference (TD) methods. Our SR-based reward function leverages this recursive approach, allowing the reward network

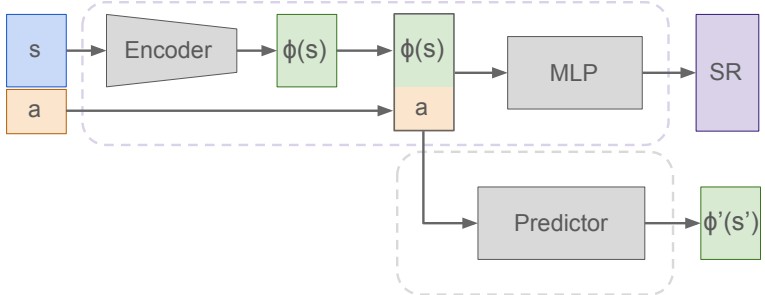

Figure 2: The architecture of the SR networks. The output of the *Encoder* is concatenated with the action to produce $\phi(s,a) = \left(\begin{smallmatrix} \phi(s) \\ a \end{smallmatrix}\right)$ in Equation 1. The result passes through a fully connected network (*MLP*) to create the $SR(s,a)$ vector. The *Predictor* network is used for an auxiliary task to help train the *Encoder*. It predicts $\phi'(s')$, an estimate of the true encoded next state $\phi(s')$, from $\left(\begin{smallmatrix} \phi(s) \\ a \end{smallmatrix}\right)$.

to be trained in parallel with the actor and critic networks, with minimal changes to the existing training pipeline.

However, directly estimating SR using these formulations becomes computationally intractable as the number of states and actions increases, or when transitioning from discrete to continuous domains. To address this, previous research (Kulkarni et al., 2016; Machado et al., 2020; Zhang et al., 2017) has extended SR to continuous state and action spaces using Successor Features Representation (SF). SF is expressed in terms of state and action features vector $\phi(s,a)$:

$$M(s_t, a_t) = \phi(s_t, a_t) + \gamma \mathbb{E}\left[M(s_{t+1}, a_{t+1})\right].\tag{1}$$

Here $M(s_t, a_t)$ is the expected future occupancy of the state-action features for $(s_t, a_t)$ where each element of this vector can be seen as tracking the SR for a single feature of the state-action space. The choice of feature extractor $\phi$ is a design decision that depends on the environment. Most existing work focuses on extracting features only from the state, not the actions. In this scenario, $\phi(s,a)$ can be represented as $\left(\begin{smallmatrix} \phi(s) \\ a \end{smallmatrix}\right)$, which is a concatenation of state features and actions. In this work, we adopt this approach and use a feature extractor network to derive features from the state only.

## 2.3 Imitation Learning via Distribution Matching

Methods like behavioral cloning (BC), which directly learn a policy $\pi(a|s)$ mapping states to actions, are straightforward and effective when ample data is available. However, they are prone to distribution shift because they only match the observed action distribution (Ross et al., 2010). During inference, as the distribution of encountered states deviates from those seen during training, the accuracy of action predictions diminishes. This leads to accumulating errors that the policy cannot correct.

Distribution matching methods, and related approaches (Ho & Ermon, 2016; Fu et al., 2018; Nachum et al., 2019; Ghasemipour et al., 2019; Ke et al., 2020; Kostrikov et al., 2020), are more robust to distribution shifts since they aim to match both the state and action distributions encountered during training. This helps keep the policy close to the states observed in demonstrations.

Formally, the occupancy measure of a state-action pair under policy $\pi$ can be defined as

$$\rho^\pi(s,a) = \mathbb{E}_\pi\left[\sum_{t=0}^\infty \gamma^t \mathbb{I}(s_t = s, a_t = a)\right],$$

where $\mathbb{I}$ is the indicator function, which equals one if the condition is met and zero otherwise. This is closely related to the state-action distribution $d^\pi(s,a) = (1-\gamma)\rho^\pi(s,a)$. Considering that there is a one-to-one correspondence between the state-action distribution and the policy (Puterman, 1994), distribution matching methods aim to indirectly learn a policy by minimizing the divergence between $d^{Expert}$ and $d^\pi$. A common

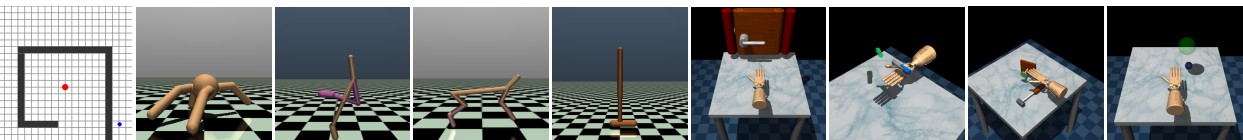

Figure 3: Environments used for our experiments. From left to right: 2D Toy Maze, MuJoCo environments: [Ant, Walker2D, HalfCheetah, Hopper], Adroit Hand environments: [Door, Pen, Hammer, Relocate]. We use the states provided by the environments in our experiments.

choice is KL-Divergence, and minimizing $D_{KL}(d^\pi || d^{Expert})$ can be viewed as maximizing the following RL objective where the reward is given by the log ratio of the state-action distributions between the expert policy and the learned policy $\pi$ (Kostrikov et al., 2020).

$$\mathbb{E}_\pi \left[ \sum_{t=0}^{\infty} \gamma^t \log \frac{d^{Expert}(s,a)}{d^\pi(s,a)} \right],$$

Since the state-action distribution is often unavailable, efforts are typically focused on estimating the ratio of the two distributions (Ho & Ermon, 2016; Nachum et al., 2019).

In this paper, we propose a method that estimates SR as a proxy for the expert's state-action distribution from demonstrations and uses it as a reward for downstream RL algorithms.

### 2.3.1 Relationship between SR and State-Action Visitation

SR implicitly captures the state-action visitation. Many density-based IL methods, such as GAIL (Ho & Ermon, 2016), use state-action distribution or occupancy measure for their distribution matching techniques. In Appendix:C, we derive the following close relationship between the occupancy measure and the successor representation

$$\rho(s',a') = \mathbb{E}_{s_0 \sim \mu_0, s \sim \mathcal{T}, a \sim \pi} \left[ M(s,a,s',a') \right].$$

The occupancy measure $\rho(s',a')$ can be seen as the expectation of successor representations $M(s,a,s',a')$ with respect to the probability of all state-action pairs $(s,a)$ that preceded $(s',a')$. We learn successor features representation (SF) from the expert demonstrations and use it as a proxy for the expert's occupancy measure.

## 3 SR-Reward

### 3.1 Architecture

We use the architecture shown in Figure 2 to estimate the SR vector in continuous state and action settings. Our architecture is built upon the works of Machado et al. (2020), Kulkarni et al. (2016), and Borsa et al. (2019) with a few notable changes. First, our SR network extends the previous works to include the action when estimating the SR. This is important as our SR-based reward function $r(s,a)$ is a function of both the state and the action and needs to distinguish the reward values of different actions. Second, it is common to use an auxiliary task when learning the encoder from scratch. Kulkarni et al. (2016) use the reconstruction of the state as the auxiliary task, while Machado et al. (2020) opt for a prediction task in which the next state is predicted from the encoded state and the action. Inspired by the results of Ni et al. (2024), we use the prediction of the next encoded state as our auxiliary task. Given the encoding of the current state $\phi(s)$ and its corresponding action in the dataset $a$, we predict the encoded next state $\phi(s')$. We use the $l^2$ loss for this auxiliary task. Finally, our encoder consists of fully connected layers with ReLU activation layer as the final layer. We normalize the feature vector to ensure that all features are in the same range, such that $\|\phi(s)\|_1 = 1$ as suggested by Machado et al. (2020). If the environment dynamics are not fully Markovian one can use a history of states as $s$ and replace the fully connected layers of the encoder with LSTM layers as proposed by Borsa et al. (2019).

## 3.2   From SR Vector to Scalar Reward

Machado et al. (2020) shows that the norm of SR implicitly counts the state visitation. Motivated by this result, we use the $l^2$-norm of the SR vector as our reward function. Intuitively, each element $i$ of the SR vector, estimated using Equation 1, is the expected discounted sum of feature $i$ of the state according to the policy that created the demonstration dataset. Hence aggregating all the elements of the SR vector in our offline setting can be seen as a visitation count of the state-action pairs when following the demonstration policy. If the demonstrations are created by an expert, $\|SR(s,a)\|_2$ represents how often the expert has visited $(s,a)$ while performing a task. Taken as the reward for offline RL, we set out to find a policy that maximizes the state-action visitation of the expert. We empirically show that we can learn competitive policies using this reward function.

## 3.3   Negative Sampling

Neural networks tend to overestimate the value of out-of-distribution data points (Thrun & Schwartz, 1999; Fujimoto et al., 2018; 2019; Ball et al., 2023). The overestimation error is especially concerning in our setup because an overestimated value of the reward for unseen states and actions will encourage the value networks and subsequently the policy to diverge from the expert demonstrations. Motivated by the idea of conservative value function via negative sampling Luo et al. (2020), we develop our negative sampling strategy to combat the overestimation error of our SR network. Similar to Luo et al. (2020) we create our negative samples $\hat{s}$ and $\hat{a}$ by adding a small Gaussian noise to states and actions from our expert trajectories. However, instead of subtracting the $l^2$-norm of the difference vector $\|s - \hat{s}\|_2$ from the reward estimate of the negative samples, we decay the values using a Gaussian, $\exp\left(-\frac{\|s-\hat{s}\|_2}{\sigma^2}\right)$, with $\sigma$ controlling the strength of the decay. Furthermore, we apply negative

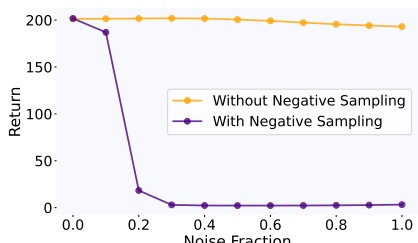

Figure 4: Mean return of corrupted expert trajectories for Relocate environment. Negative sampling significantly reduces the reward for states and actions further away from the expert demonstrations.

sampling not to the space of value functions but to the space of rewards. This is possible in our setting where a reward function is estimated and used for learning a policy. Having control over the reward function in this setting provides the opportunity to build conservatism directly into the value functions and subsequently the policy by modifying the reward instead of forcing the value function or the policy to act conservatively Fujimoto et al. (2018; 2019); Kumar et al. (2020).

Our negative sampling strategy is effective primarily within the local vicinity of expert demonstrations, achieved by introducing perturbations to expert trajectories. Therefore, it does not prevent reward overestimation for state-action pairs that significantly diverge from the expert's behavior. As a result, the SR-Reward is more suitable for offline settings, where exploration is limited. In online settings, where the agent can encounter unfamiliar state-action pairs with overestimated rewards, this could lead to suboptimal policy learning.

Figure 5 shows the effect of our negative sampling strategy on a toy environment. We train an SR network using ten demonstrations with and without negative sampling and evaluate the rewards over the grid space for each one of the cardinal directions. The mean value plot in Figure 5 shows how negative sampling during training prevents overestimation error for the rewards of the state-action pairs not seen in the demonstrations. Plots for the four main directions show higher reward estimates for movement in the corresponding direction. For example, the `Left` plot shows the reward for moving left at every grid point and as expected this reward is higher for the upper portion of the trajectories where the expert has moved left.

To highlight the impact of negative sampling in mitigating overestimation errors, Figure 4 compares the mean trajectory returns of models trained with and without negative sampling. Expert trajectories are corrupted by varying levels of Gaussian noise, and their episodic returns are estimated using SR-Reward. The model trained with negative sampling exhibits significantly lower returns for corrupted trajectories, reflecting its enhanced sensitivity to out-of-distribution data. This effect is evident in the initial drop in returns when

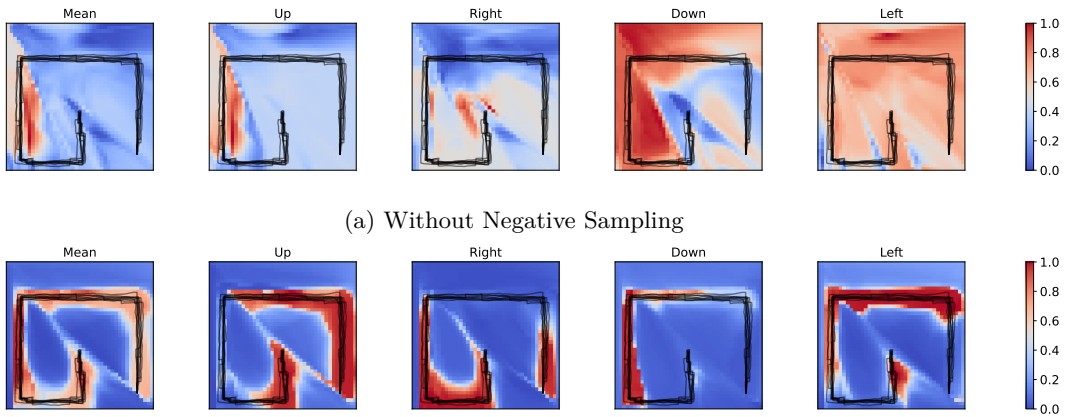

(a) Without Negative Sampling

(b) With Negative Sampling

Figure 5: The plots show the effect using negative sampling for a 2D Toy Maze environment (Figure 3)with continuous states and actions. The black lines represent the trajectories of the expert starting near the bottom-right and moving counter-clockwise towards the goal near the center. The mean of the SR-Reward over four directions [Up, Right, Down, Left] and the SR-Reward associated with each direction is plotted. Using negative sampling significantly reduces the extrapolation error for out-of-distribution state-action pairs.

Gaussian noise with $\sigma = 0.1$ is applied, followed by a much steeper decline as the noise level increases. In contrast, the model trained without negative sampling produces similar reward estimates for both expert and corrupted trajectories, with returns showing only a modest decline under substantial noise levels. Different environments may demand varying degrees of sensitivity to noise and out-of-distribution data. Our negative sampling strategy offers flexibility, allowing customization for each environment by adjusting the perturbation noise applied to expert trajectories and fine-tuning $\sigma$ for the decay rate in Equation 2.

## 3.4   Training

We employ several loss functions to train our SR network. As mentioned in Section 2.2, we can estimate the SR using the Bellman equation in a continuous state-action setting. The reward for the Bellman target in Equation 1 is replaced with $\phi(s, a) = \left(\begin{smallmatrix} \phi(s) \\ a \end{smallmatrix}\right)$ which is the concatenation of the encoded state and the action. We use the $l^2$-loss to minimize the Bellman error:

$$\mathcal{L}_{Bellman} = \mathbb{E}_{(s,a,s',a') \sim \mathcal{D}} \left[ (M(s,a) - (\phi(s,a) + \gamma M(s',a')))^2 \right]$$

Auxiliary tasks—such as predicting the reward, next state, or next latent state—have been shown to facilitate the learning of useful latent representations for downstream tasks Fujimoto et al. (2024); Ni et al. (2024). To support encoder training, we incorporate an auxiliary prediction task in which the model learns to predict the next encoded state $\phi(s')$ from the current encoded state $\phi(s)$ and action $a$. We compute the $l^2$-loss as

$$\mathcal{L}_{Prediction} = \mathbb{E}_{(s,a,s') \sim \mathcal{D}} \left[ (\phi(s') - Predictor(\phi(s), a))^2 \right]$$

Ensuring that the reward is bounded is crucial for the stability of reinforcement learning algorithms. Prior works have addressed this in various ways, such as clipping observed rewards or adding a penalty on the magnitude of the estimated reward Mnih et al. (2015); Garg et al. (2021). In our approach, we introduce an additional loss term that penalizes reward magnitudes exceeding 1, effectively encouraging a soft upper bound on the reward. This regularization was found to improve training stability. As described in Section 3.2, we define the reward as the $l^2$-norm of the SR vector.

$$\mathcal{L}_{Magnitude} = \mathbb{E}_{(s,a) \sim \mathcal{D}} \left[ \max(Reward(s,a) - 1, 0)^2 \right]$$

---

**Algorithm 1** SR-Reward + RL

---

1: **Given**: $\mathcal{D} : [(s, a, s', a')^i]_{i=0}^N, \gamma, \beta, \sigma$
2: **Initialize**: Encoder: **Enc**, Fully Connected Block: **MLP**, Predictor: **Pred**
3: **for** each training step **do**
4:      Sample $(s, a, s', a') \sim \mathcal{D}$
5:      $\phi(s) \leftarrow Enc(s)$
6:      $\phi(s') \leftarrow Enc(s')$
7:      $SR \leftarrow MLP(\phi(s), a)$
8:      $SR_{target} \leftarrow \left( {\phi(s) \atop a} \right) + \gamma MLP(\phi(s'), a')$
9:      $\mathcal{L}_{Bellman} \leftarrow MSE(SR, SR_{target})$
10:      $\phi_{pred}(s') \leftarrow Pred(\phi(s), a)$
11:      $\mathcal{L}_{Prediction} \leftarrow MSE(\phi_{pred}(s'), \phi(s'))$
12:      $r \leftarrow \|SR\|_2$
13:      $\mathcal{L}_{Magnitude} \leftarrow (max(r - 1, 0))^2$
14:      $\tilde{s} \leftarrow s + \mathcal{N}(0, \beta)$
15:      $\tilde{a} \leftarrow a + \mathcal{N}(0, \beta)$
16:      $\phi(\tilde{s}) \leftarrow Enc(\tilde{s})$
17:      $\alpha_{decay} \leftarrow exp(\frac{-\|\left( {\phi(s) \atop a} \right) - \left( {\phi(\tilde{s}) \atop \tilde{a}} \right)\|_2}{\sigma^2})$
18:      $\tilde{SR} \leftarrow MLP(\phi(\tilde{s}), \tilde{a})$
19:      $\tilde{r} \leftarrow \|\tilde{SR}\|_2$
20:      $\mathcal{L}_{Neg.Sample} \leftarrow MSE(\tilde{r}, \alpha_{decay} \times r)$
21:      $\mathcal{L}_{Total} \leftarrow \mathcal{L}_{Bellman} + \mathcal{L}_{Prediction} + \mathcal{L}_{Magnitude} + \mathcal{L}_{Neg.Sample}$
22:      $s \leftarrow \left( {s \atop \tilde{s}} \right), a \leftarrow \left( {a \atop \tilde{a}} \right), s' \leftarrow \left( {s' \atop s'} \right), r \leftarrow \left( {r \atop \tilde{r}} \right)$
23:      $RL(s, a, r, s')$
24: **end for**

---

Finally, we add a negative sampling loss to improve the robustness of the reward function for out-of-distribution state-action pairs. Similar to Luo et al. (2020) we create negative samples $\tilde{s}$ and $\tilde{a}$ by perturbing states and actions from the demonstrations with noise. While one might expect negative samples to overlap with the distribution of demonstrations and affect the estimation of the SR, prior work Luo et al. (2020) shows that demonstrations typically cover only a small subset of the state-action space. As a result, negative samples are highly likely to be orthogonal to the demonstrations, an effect that becomes more pronounced as the dimensionality of the environment increases. We use isotropic Gaussian noise $\mathcal{N}(0, \beta)$ to create the negative samples. The hyperparameter $\beta$ controls the standard deviation of the Gaussian noise. Intuitively, we want perturbed state-action pairs $(\tilde{s}, \tilde{a})$ to have lower reward values proportional to the distance from their counterpart $(s, a)$ from the dataset. Since SR estimates the visitation count based on $\phi(s, a) = \left( {\phi(s) \atop a} \right)$, we measure the distance between the negative samples and their original counterparts in the space of features and actions $\left( {\phi(s) \atop a} \right)$. We calculate the decay factor using an exponential kernel as

$$\alpha_{decay} = \exp\left( \frac{-\|\phi(s, a) - \phi(\tilde{s}, \tilde{a})\|_2}{\sigma^2} \right). \tag{2}$$

$\sigma$ can also be adjusted as a hyperparameter. Higher values of $\sigma$ will produce a softer decay for the reward of negative samples. The $l^2$-loss is used to correct the estimation of SR for negative samples:

$$\mathcal{L}_{Neg.Sample} = \mathbb{E}_{(s,a) \sim \mathcal{D}} \left[ (Reward(\tilde{s}, \tilde{a}) - \alpha_{decay} \times Reward(s, a))^2 \right]$$

We train our SR network using the summation of all losses as our total loss:

$$\mathcal{L}_{Total} = \mathcal{L}_{Bellman} + \mathcal{L}_{Prediction} + \mathcal{L}_{Magnitude} + \mathcal{L}_{Neg.Sample}$$

Algorithm 1 shows the pseudocode for training the SR-Reward and the offline RL in the same loop. The sampled transitions used for training the SR networks have the form $(s, a, s', a')$ which is different from the

Table 1: Normalized mean return and standard deviation over five seeds from different algorithms trained on D4RL datasets (Fu et al., 2020). Offline RL algorithms using SR-Reward perform similarly to the ones using the true reward from the environment. Maniskill2 datasets Gu et al. (2023) (PickCube, StackCube, TurnFaucet) do not contain rewards and so are only compared to BC.

| | | f-DVL | | sparseQL | |
|---|---|---|---|---|---|
| **Env** | **BC** | True Reward | SR-Reward (Ours) | True Reward | SR-Reward (Ours) |
| Ant | 86.33 ± 3.91 | 84.35±4.34 | 82.75±4.92 | **86.64±5.14** | 82.25±7.09 |
| Hopper | 108.73 ± 4.39 | **110.74±1.14** | 108.25±3.16 | 110.32±2.69 | 109.59±0.52 |
| Halfcheetah | 104.87 ± 1.55 | 104.70±0.36 | 103.80±1.99 | **106.23±1.38** | 105.96 ± 0.82 |
| Walker2d | 71.39± 14.95 | **84.41±14.49** | 78.48±5.52 | 84.02±13.75 | 74.48 ± 7.85 |
| Door | 76.93 ± 21.81 | 96.97 ± 4.70 | 98.40 ± 3.33 | 78.28 ± 23.98 | **104.10 ± 1.57** |
| Hammer | 114.30 ± 4.59 | 90.53 ± 10.28 | 111.52 ± 9.74 | 71.32 ± 18.50 | **117.17 ± 3.71** |
| Pen | 104.95 ± 5.70 | **109.67 ± 10.06** | 97.79 ± 5.15 | 108.75 ± 3.91 | 105.00 ± 3.57 |
| Relocate | **93.20 ± 4.15** | 92.36 ± 4.35 | 88.22 ± 7.30 | 80.69 ± 8.15 | 90.91 ± 6.99 |
| PickCube | 91.79 ± 4.23 | — | 88.81 ± 5.72 | — | **96.70 ± 2.21** |
| StackCube | 51.41 ± 3.97 | — | 70.71 ± 12.55 | — | **71.03 ± 18.41** |
| TurnFaucet | 20.26 ± 1.27 | — | 35.44 ± 6.83 | — | **51.64 ± 9.41** |

ones typically used for RL due to the addition of the next action $a'$. This form of transition, however, can be easily produced with access to a set of demonstrations $\mathcal{D}$. We warm-start the training loop by pre-training the SR networks for 10,000 steps before using its SR-Reward to train the RL agent.

## 4 Experimental Evaluation

### 4.1 Setup

To evaluate the proposed reward module, we integrate it with two distinct offline RL algorithms: f-DVL (Sikchi et al., 2023) and SparseQL (Xu et al., 2023). Both algorithms, which build on foundational concepts from IQL (Kostrikov et al., 2022) and XQL (Garg et al., 2023), have demonstrated enhanced stability and strong performance in offline reinforcement learning settings. In our experiments, we replace the rewards in the offline dataset with those generated by SR-Reward, allowing the reward function to be learned in conjunction with the RL algorithms.

For empirical validation, we utilize the widely-used MuJoCo-based (Todorov et al., 2012) environments for locomotion tasks, and the Adroit hand (Rajeswaran et al., 2018) environments for assessing performance on more realistic, hand-engineered rewards. Figure 3 illustrates these environments. We also include three challenging environments from Maniskill2 (Gu et al., 2023), shown in Figure 6. These environments pose greater difficulty than MuJoCo and Adroit due to variable start and goal positions, extended time horizons, and the complexity of manipulating objects with only two fingers. As in

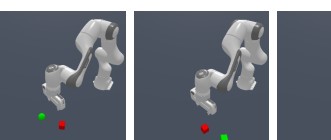

Figure 6: PickCube (Left), StackCube (Middle) and TurnFaucet (Right) from Maniskill2 (Gu et al., 2023)

real-world applications, these datasets do not include rewards, prohibiting the use of offline RL algorithms. We use SR-Reward in combination with SparseQL as the RL agent, for Maniskill2 environments as well as in ablation studies for data size, data quality, and negative sampling (Appendix A, B, and G).

We use state information from the environment and datasets, and hence do not use an image encoder for our experiments. Each agent is trained for one million gradient steps (two million gradient steps for ManiSkill environments), with five different random seeds used per task. For each training run (across five random seeds), we save the checkpoint that achieves the highest mean return over 25 evaluation rollouts, measured every 10,000 steps during training. After selecting the best checkpoint for each seed, we then evaluate it

on 50 fresh rollouts and report the mean and standard deviation across the five seeds (one mean return per seed). Table 1 shows the mean return and standard deviation over five seeds for different environments.

Hyperparameters remain largely consistent across environments, with key parameters listed in Table 2 (Appendix D). Further discussion on the hyperparameters of the negative sampling strategy and how to choose them is presented in Appendix G. We use the offline datasets from D4RL (Fu et al., 2020), and follow their normalization procedures, employing the provided scores for random and expert demonstrators (Appendix E).

Our experiments aim to answer the following key questions:

1. Can SR-Reward effectively replace the true reward signal for offline RL? (4.2)

2. How does the performance of offline RL algorithms using SR-Reward compare to that of BC (4.3)

3. Is negative sampling strategy a necessary component of SR-Reward? (4.4)

### 4.2 Question 1: Can SR-Reward replace the true reward signal?

To address this question, we use the true reward signal from the environment as a baseline and compare the performance of offline RL algorithms (f-DVL and sparseQL) using the true reward against the reward signal generated by SR-Reward.

The MuJoCo and Adroit Hand environments differ significantly in the complexity of their reward functions. In MuJoCo environments, the objective is relatively simple: the agent must move to the right as quickly as possible. A straightforward reward function based on the agent's velocity suffices for such tasks. In contrast, Adroit Hand environments present more intricate challenges, such as rotating a pen with one hand or hammering a nail. These tasks require dense reward functions, which are constructed using multiple carefully designed sub-rewards and thresholding.

Table 1 summarizes the performance of the offline algorithms using SR-Reward and the true reward. For the MuJoCo environments, the performance achieved with SR-Reward closely matches that of the true reward, demonstrating that the dense reward generated by SR-Reward is as informative as the environment-provided reward. Notably, in the Adroit Hand environments, SR-Reward yields similar or even higher performance than the true reward. This result highlights the inherent challenges of manually engineering reward functions for complex tasks and underscores the advantages of using SR-Reward, particularly for scenarios where crafting a reward function is not straightforward.

Furthermore, Figure 8 and Figure 9 present ablation studies on dataset size and quality, demonstrating that SR-Reward can produce an informative reward signal and sustain competitive performance even under limited or degraded data conditions. While overall performance decreases as the dataset size is reduced or the data quality deteriorates, this trend affects all algorithms similarly. Notably, RL agents (SparseQL) trained with SR-Reward maintain performance levels comparable to those trained with the true environment reward or to behavioral cloning agents, even when learning from suboptimal data. (See Appendix A and Appendix B for further discussion.)

### 4.3 Question2: How does SR-Reward + offline RL perform compared to BC?

Behavioral Cloning (BC) is a straightforward and effective method in scenarios where no reward function exists, but ample demonstrations are available. To highlight the advantages of using offline RL equipped with SR-Reward, we compare its performance against BC across MuJoCo, Adroit Hand, and more complex ManiSkill2 robot manipulation tasks. In the latter, no true reward signal is available in the datasets, and the offline RL algorithms rely solely on SR-Reward for reward generation.

As shown in Table 1, combining SR-Reward with offline RL performs on par with BC for MuJoCo and Adroit Hand environments. However, as task complexity increases, the performance gap widens. This trend is particularly evident in the ManiSkill2 environments (Figure 7 (`Left`)), where BC closely matches the performance of SR-Reward-equipped sparseQL for the relatively simple PickCube task but falls behind in

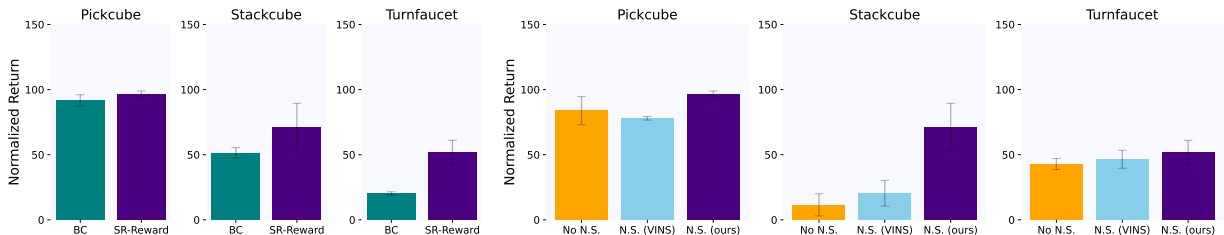

Figure 7: Performance on Maniskill2 environments. Turning the faucet requires a continuous connection with the edge of the handlebar throughout the movement, and stacking the cubes requires more precision than simply relocating them. SR-Reward + RL (sparseQL) outperforms BC as the difficulty of the task increases (Left). The benefits of using Negative sampling (N.S.) become more prominent for more difficult tasks (Right). Our negative sampling based on exponential kernel outperforms that of Luo et al. (2020) (VINS), especially on the StackCube task.

the more demanding StackCube and TurnFaucet scenarios, which require higher precision for successful completion. Similar results hold when combining f-DVL with SR-Reward, although BC shows slightly better performance on a the simpler PickCube task but falls behind on StackCube and TurnFaucet.

We have limited our experiments to two offline RL algorithms (f-DVL and SparseQL), however, our findings underscore the versatility of SR-Reward as a modular component that can, in principle, be compatible with many offline RL algorithms by replacing the reward function. This flexibility makes it well-suited for applications in robotics and other domains where obtaining demonstrations is more feasible than designing dense reward functions, and where RL algorithms have the potential to surpass the limitations of BC.

### 4.4 Question3: Is negative sampling strategy a necessary component of SR-Reward?

A narrow subset of the overall state-action space covered by the demonstrations can lead function approximation networks to predict erroneously high values when queried for out-of-distribution data. Intuitively, the negative sampling strategy presented in this paper can be beneficial in increasing the robustness of the networks used in SR-Reward by lowering the resulting reward values for state-action pairs outside of demonstrations. To evaluate its effectiveness, we train SR-Reward both with and without the negative sampling strategy and compare the results in the PickCube, StackCube, and TurnFaucet environments (Figure 6).

As shown in Figure 7 (`Right`), the use of the negative sampling strategy leads to similar or improved performance across all tasks. Additionally, we benchmark this approach against the linear decay strategy proposed by Luo et al. (2020), applied to SR-Reward (rather than the value function). Our results demonstrate that the exponential decay kernel used in our negative sampling strategy yields better performance. While the negative sampling strategy consistently enhances results across tasks, the magnitude of its contribution varies depending on the task.

## 5 Related Work

Learning to perform a task from offline data has been extensively studied under the IL and IRL umbrella (Abbeel & Ng, 2004; Ho & Ermon, 2016; Fu et al., 2018; Garg et al., 2021; Kostrikov et al., 2020; Kalweit et al., 2020; Pomerleau, 1991). One common approach is methods based on behavioral cloning (Pomerleau, 1991) which reduce imitation learning to a supervised learning problem, i.e., learning a mapping from environment states to expert actions. They aim to increase the probability of expert actions for the states seen in the demonstrations. Although this approach can work in simple environments with large amounts of data, it is inherently myopic and fails to reason about the consequences of its selected actions. Consequently, such greedy approaches suffer from compounding errors due to covariant shift (Ross et al., 2010) when the agent deviates from the demonstrated states.

In contrast, IRL methods incorporate information about the environment dynamics into the decision-making process by imitating the expert actions as well as the visited states (Abbeel & Ng, 2004). Many IRL methods, such as GAIL (Ho & Ermon, 2016) and its extensions, simultaneously estimate the reward function that best explains the expert behavior and its associated policy. This optimization is done using an adversarial scheme with the discriminator trying to distinguish between the expert trajectories and ones generated by the learned policy. Simultaneously, the discriminator's error is used as the reward signal for training the policy. The adversarial nature of the training strategy makes such algorithms prone to training instabilities(Goodfellow et al., 2014; Kostrikov et al., 2019). Additionally, they require further interactions with the environment during training to create a dataset of non-expert trajectories for training the discriminator. Furthermore, there are no theoretical guarantees that show adversarial training to lead to a better performance than a two-step process, which first infers a reward function from demonstrations, followed by learning a policy using the previously inferred reward (Liu et al., 2021). In this work, we part from adversarial training and so decouple the learning process of reward function and policy while training both simultaneously.

Already there have been efforts in bypassing the adversarial optimization. Kalweit et al. (2020) derived an analytical solution for the reward based on the assumption that expert policy follows a Boltzmann distribution. Their formulation applies to continuous states but is limited to discrete action spaces. While sharing a similar objective to ValueDICE (Kostrikov et al., 2020), Garg et al. (2021) removes the need for adversarial training by formulating the reward function in terms of the value functions and maximizing them. Their objective function implicitly reduces a distance measure, such as $\chi^2$-divergence, between the occupancy measure of the expert and the one of the policy being trained. This approach does not yield an explicit reward model, but a reward value can be extracted from the learned value function and policy. Without optimizing for the optimal reward function Reddy et al. (2020) uses a simple binary indicator as the reward which distinguishes between expert demonstrations and online interactions. Our reward function can be seen as the continuous version of SQIL (Reddy et al., 2020) because the SR value of states and actions that are visited by the expert, will be naturally higher than the rest.

The growing need to fine-tune large language models has led to a surge of research focused on learning reward functions from human feedback and using them for model optimization (Christiano et al., 2023; Rafailov et al., 2023; Ethayarajh et al., 2024). These approaches typically assume a Bradley-Terry model (Bradley & Terry, 1952) to represent the reward function and rely on datasets containing both positive and negative feedback. In contrast, our work focuses on learning a reward function solely from expert demonstrations, without requiring any form of negative feedback. There is also a series of works that modify the existing environment reward to gain improvements during training. Vieillard et al. (2020) suggests adding $\log(\pi(a|s))$ of the policy $\pi$ that is being learned to the reward in temporal difference (TD) learning. The authors argue that the logarithm of the policy is a strong learning signal as it is available even in a sparse reward setting and since its value is close to zero for optimal actions under optimal policy this does not conflict with the optimal control objective.

Recent works have aimed to use successor representation in reinforcement learning (Barreto et al., 2017; Zhang et al., 2017; Filos et al., 2021; Brantley et al., 2021; Jain et al., 2024). While Barreto et al. (2017) and Zhang et al. (2017) use SR to generalize the value function to different rewards for transfer learning, Brantley et al. (2021) focus on generalizing the representation of SR over policies for small partially observable environments with known dynamics. In multi-agent settings Filos et al. (2021) learn the shared features of the environment using the estimated SR of all other agents irrespective of their goals. Jain et al. (2024) uses SR in the context of learning from demonstrations by matching the SR of the learner's policy to that of the expert. We share the same motivation for our work but we use SR as a reward function and do not require online interactions during training. Other works such as (Moskovitz et al., 2022; Machado et al., 2020) modify the reward using the SR of the policy that is being learned. Machado et al. (2020) showed that the norm of the SR vector can act as the proxy for the state visitation count. They modify the reward from the environment by adding the inverse of this state visitation count during training. Moskovitz et al. (2022) suggest a modification to SR to only consider the first visitation of a state, hence learning the expected discounted time to reach successor states. Similar to Machado et al. (2020), they also make use of the inverse of their modified SR norm and show improved performance, especially in scenarios where the reward in a given state will be depleted after the first visit.

Our work is similar to the one of Machado et al. (2020) in the sense that we are also viewing the norm of SR vector as a proxy to state visitation count. However, we are working in an offline IRL setting where there is no other reward available and the dataset is fixed. We show that in the absence of a reward signal from the environment, one can use the norm of the SR vector directly as the reward. Additionally, we extend the SR vector to continuous actions and employ a negative sampling procedure to lower the value of our SR-based reward for state-action pairs in the vicinity of the demonstrations that were not present in the demonstrations dataset, hence combating the extrapolation error and creating a more robust reward function for offline RL algorithms.

## 6 Conclusion

We introduced SR-Reward, a reward function based on successor representation, which is learned from offline expert demonstrations. This reward function assigns high rewards to state-action pairs frequently visited by expert demonstrators. SR-Reward is independent of both policy and value functions but can be trained concurrently with them, enabling easy integration with various RL algorithms without requiring significant modifications to the training pipeline.

Additionally, we implemented a negative sampling strategy to encourage a pessimistic estimation of rewards for out-of-distribution state-action pairs, thereby making the reward function more resistant to overestimation errors. Our empirical results demonstrate that SR-Reward can effectively serve as a proxy for the true reward in scenarios where no reward function is available or where the complexity of the task makes it difficult to hand-engineer sufficiently informative reward functions.

## 7 Limitations and Future Work

We focused our experiments on offline settings because the negative sampling strategy can only protect the SR-Reward from overestimation errors near the expert demonstrations, where meaningful negative samples are generated by perturbing expert trajectories. Since expert trajectories cover only a small portion of the state space, high extrapolation errors can be expected in regions far from these demonstrations. Consequently, in online RL, when the agent explores areas distant from the expert trajectories, it may be misled by inflated rewards, leading to the learning of suboptimal policies.

SR-Reward assumes the availability of a dataset of optimal trajectories. Although our empirical results indicate a degree of robustness when combining optimal and sub-optimal datasets (Figure 9), the presence of sub-optimal demonstrations can negatively impact SR-Reward since the training process treats optimal and sub-optimal demonstrations equally. Enhancing the ability to control the influence of demonstrations based on their quality could lead to higher-quality rewards and more data-efficient learning, offering a promising direction for future research.

Given that the successor representation is closely linked to occupancy measures and state-action distributions, the SR-Reward function proposed here can be employed to approximate the state-action distributions of both expert and non-expert actors. This paves the way for developing new algorithms in imitation learning (IL) and inverse reinforcement learning (IRL), enabling the direct matching of distributions using an approximate model of state-action distributions. We consider this an exciting direction for further exploration and future research.

Finally, a theoretical investigation regarding the convergence properties of using SR as a reward or the efficacy of using L2 Norm for converting the SR vector to a scalar reward can provide more support for our claim, however, in this paper, we have focused on supporting our claims using empirical results.

## 8 Acknowledgment

This work was funded by Carl Zeiss Foundation throught the ReScaLe project.

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

# A    Data-Size Ablation

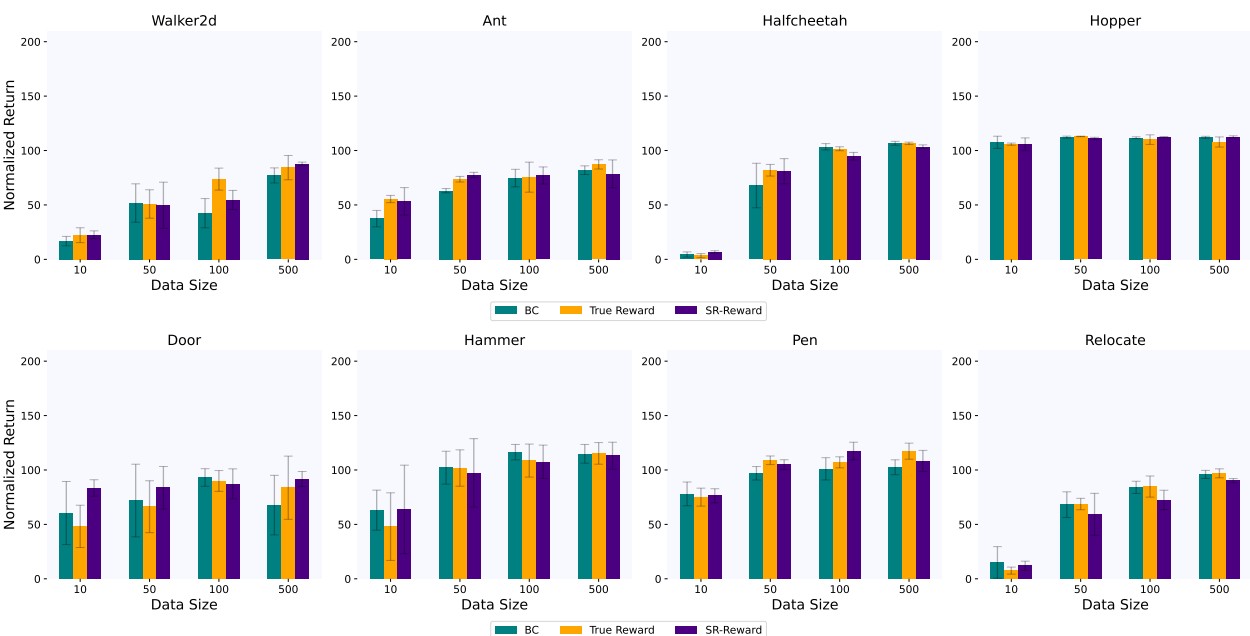

Figure 8: Effect of data size on performance. RL agents (SparseQL) using SR-Reward show competitive performance compared to BC and the RL agents that use the true reward.

To investigate the impact of data size on SR-Reward, we trained each algorithm using varying numbers of expert trajectories from the D4RL dataset. We used sparseQL as our offline RL algorithm. Specifically, we evaluated performance using [10, 50, 100, 500] demonstrations for MuJoCo and Adroit hand environments. As shown in Figure 8, agents trained with true reward do not significantly outperform those trained with SR-Reward across different data sizes in all MuJoCo environments.

As the number of demonstrations decreases, performance declines for all agents, regardless of the reward function used. This trend suggests that informative rewards can still be learned even with limited data. Therefore, the performance drop observed with fewer demonstrations likely reflects the data inefficiency of the offline RL algorithms rather than a significant decline in the quality of the learned reward.

# B Data-Quality Ablation

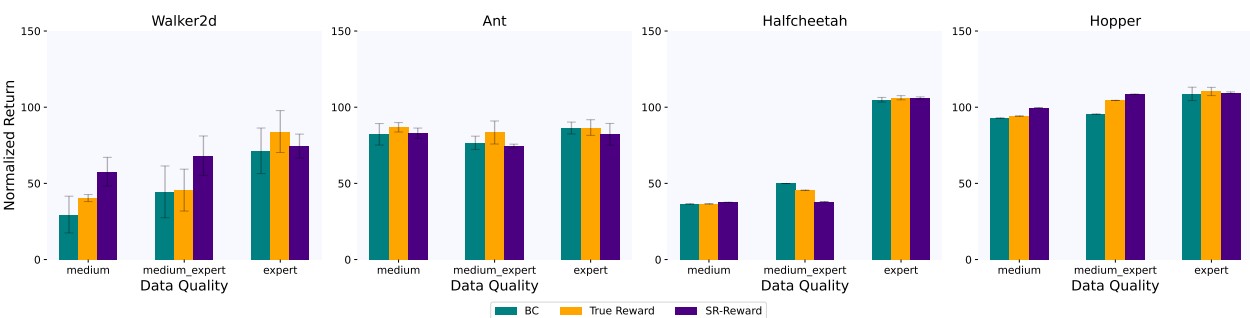

Figure 9: Effect of data quality on performance. RL agents (SparseQL) using SR-Reward show similar performance to the baselines. SR-Reward is robust to the mixing of sub-optimal demonstrations (medium-expert) as there is no significant drop in performance compared to the agents that were trained on the true reward.

Depending on the environment, creating a set of high-quality expert demonstrations can quickly become a cumbersome task. Therefore, it is important to know the effect of sub-optimal demonstrations when used for training the SR-Reward. We conduct our experiments on MuJoCo environments using three datasets with different quality demonstrations from D4RL with the "medium-expert" dataset being a combination of both expert and medium demonstrations. Figure 9 shows that agents using SR-Reward have similar performance to the ones trained using true environment reward. The mixing of expert and medium datasets does not show a significant negative impact on agents trained with SR-Reward as compared to other agents. In fact including the sub-optimal trajectories results in higher returns, especially for the more difficult Walker2D environment which can benefit from larger datasets. Having low sensitivity to sub-optimal demonstrations is a desirable attribute of SR-Reward since collecting expert demonstrations can be tedious and error-prone, which increases the possibility of including sub-optimal demonstrations.

## C    Occupancy Measure and Successor Representations

We will restrict ourselves to the occupancy measure of the state only (instead of state and action). The extension to state and action is trivial via a second summation over the actions.

The expectation is with respect to starting state distribution $\mu_0$, the policy that is followed $\pi$, and the transition dynamics of the environment $\mathcal{T}$.

We can write the definitions of occupancy measures $\rho(s)$ and successor representations $M(s, s')$ in terms of probabilities $p$.

$$
\begin{aligned}
M(s, s') &= \mathbb{E}[\sum_{t=0}^{\infty} \gamma^t \mathbb{I}(s_t = s')|s_0 = s] \\
&= \sum_{t=0}^{\infty} \gamma^t \mathbb{E}[\mathbb{I}(s_t = s')|s_0 = s] \\
&= \sum_{t=0}^{\infty} \gamma^t p(s_t = s'|s_0 = s)
\end{aligned}
$$

and similarly for the occupancy measure $\rho(s)$:

$$
\begin{aligned}
\rho(s) &= \mathbb{E}[\sum_{t=0}^{\infty} \gamma^t \mathbb{I}(s_t = s)] \\
&= \sum_{t=0}^{\infty} \gamma^t \mathbb{E}[\mathbb{I}(s_t = s)] \\
&= \sum_{t=0}^{\infty} \gamma^t p(s_t = s)
\end{aligned}
$$

Below we show that $\rho(s') = \sum_s p(s)M(s, s')$:

$$
\begin{aligned}
\rho(s') &= \sum_{t=0}^{\infty} \gamma^t p(s_t = s') \\
&= \sum_{t=0}^{\infty} \gamma^t \sum_s p(s)p(s_t = s'|s_0 = s) \\
&= \sum_s p(s) \sum_{t=0}^{\infty} \gamma^t p(s_t = s'|s_0 = s) \\
&= \sum_s p(s)M(s, s')
\end{aligned}
$$

# D    Hyperparameters

Table 2: Most important Hyperparameters used in the experiments.

| Hyperparameter | Value |
| --- | --- |
| Noise $\beta$ (MuJoCo) | 1.0 |
| Noise $\sigma$ (MuJoCo) | 3.0 |
| Noise $\beta$ (Adroit) | 0.1 |
| Noise $\sigma$ (Adroit) | 0.3 |
| Noise $\beta$ (Maniskill2) | 0.03 |
| Noise $\sigma$ (Maniskill2) | 0.3 |
| LR (Critic, Value) | 0.0003 |
| LR (Actor, SR-Reward) | 0.0001 |
| Encoder MLP | [256, 128] |
| SRNet MLP | [128] |
| Predictor MLP | [128, 32] |
| Critic MLP | [256, 256] |
| Actor MLP | [128, 128] |
| ValueNet MLP | [128, 128] |
| Batch Size | 128 |
| Training Steps | 1000000 |

# E   D4RL Return Normalization

We follow the same normalization procedure as described in D4RL with min and max scores for each task taken from the D4RL datasets as below:

Table 3: Min and Max scores for each D4RL environment

| Environment | Min | Max |
|---|---|---|
| Walker2d | 1.629 | 4592.3 |
| Ant | -325.6 | 3879.7 |
| HalfCheetah | -280.178 | 12135.0 |
| Hopper | -20.272 | 3234.3 |
| Door | -56.512 | 2880.569 |
| Hammer | -274.856 | 12794.134 |
| Pen | 96.262 | 3076.833 |
| Relocate | -6.425 | 4233.877 |

For Maniskill2 environments (PickCube, StackCube, TurnFaucet), the minimum score is considered 0 when the task is not solved, and the maximum score is:

$$\text{score}_{max} = 1.0 + (1.0 - \frac{k}{\text{MAXSTEPS}})$$

where MAXSTEPS is set to 500 and $k$ is the steps of the simulation hence giving more rewards to the successful tasks that are completed in fewer steps. The expert can complete the tasks in approximately 150 steps hence the maximum score for these environments is set to 1.7.

The returns are normalized for all plots using the Min and Max scores of each environment as follows:

$$Return_{normalized} = \frac{Return - Score_{min}}{Score_{max} - Score_{min}}$$

# F Policy Comparison in the Action Space

We visualize the actions proposed by two policies on the Adroit Door task to assess their behavioral similarity. Specifically, we compare two models trained using the f-DVL algorithm: one trained with the true reward provided by the environment, and the other trained with our learned SR-Reward. Since both models achieve comparable mean returns (see Table 1), we examine whether they also produce similar actions, which would indicate that SR-Reward leads to a policy behaviorally close to that of the true reward.

Figure 10 illustrates the actions suggested by each policy when evaluated on the same observations drawn from 50 trajectories in the offline dataset. Each subplot corresponds to one dimension of the action space. The alignment between the two sets of actions across all dimensions suggests that the policies behave similarly—supporting the idea that policies trained using SR-Reward behave similarly to the ones trained using the true reward.

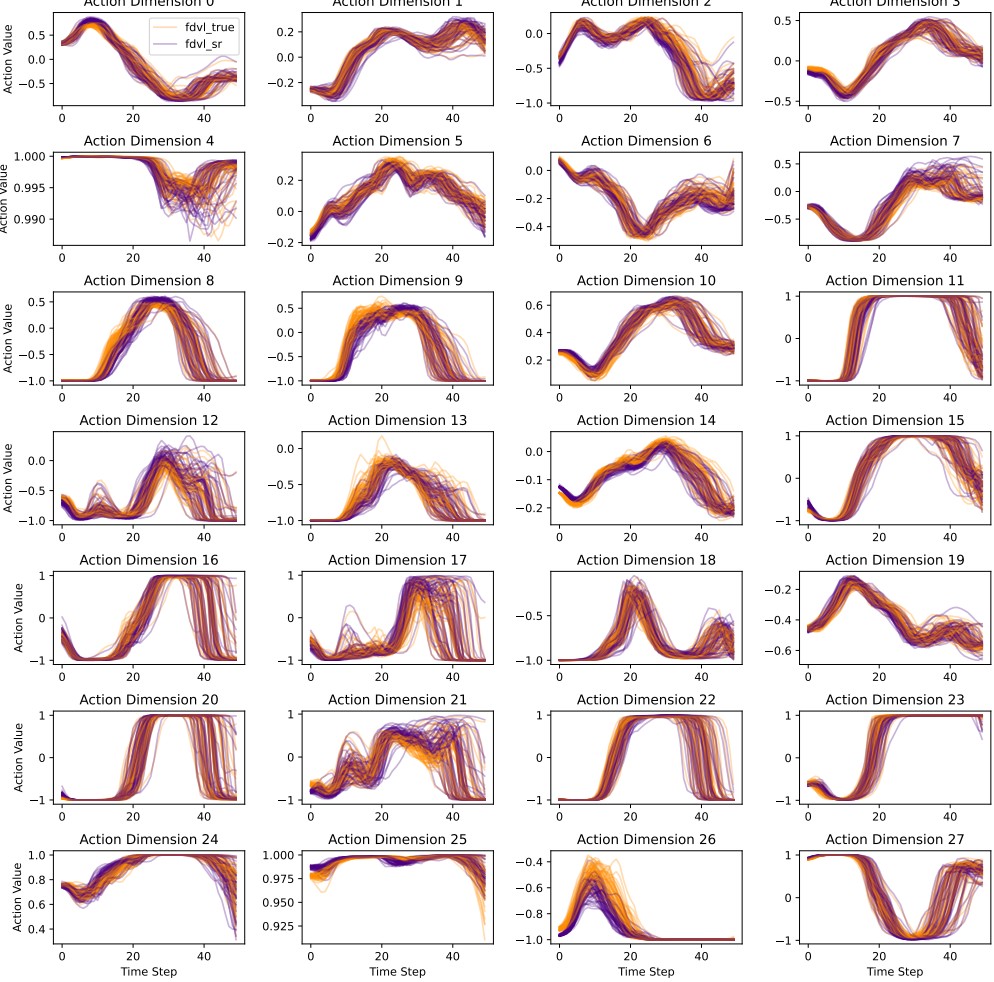

Figure 10: Action trajectories for f-DVL model trained on true reward vs SR-Reward on the Ardoit Door environment.

# G   Negative Sampling Hyperparameters

The negative sampling method introduced in Section 3.3 is governed by two hyperparameters: $\beta$ and $\sigma$. The parameter $\beta$ controls the amount of noise added to expert state-action pairs to generate negative samples, while $\sigma$ determines the width of the Gaussian penalty kernel used to measure the similarity between expert and negative samples in the latent space. To study the sensitivity of our method to these hyperparameters, we trained a series of SparseQL + SR-Reward models on the StackCube environment using a grid of different $\beta$ and $\sigma$ values. Figure 11 presents the normalized returns across these settings. The red circle marks the hyperparameter configuration used in our main experiments (Section 4).

Overall, high values of $\beta$ tend to degrade performance, likely because the resulting negative samples become out-of-distribution with respect to the demonstration data, which can negatively affect training. A similar trend is observed for large values of $\sigma$: when the penalty kernel is too wide, both expert and perturbed samples receive similar rewards, making it harder to distinguish optimal from suboptimal behaviors, ultimately harming performance. As a practical guideline, we set $\beta$ to approximately the median of the standard deviations across all observation and action dimensions in the demonstration data. Figure 12 shows the distribution of values across these dimensions, with the red line marking the median standard deviation. This choice typically yields a noise level that reflects the natural variability present in the expert data. For $\sigma$, we choose a value between 3 to 7 times the selected $\beta$. This ensures that the reward function remains sensitive enough to distinguish perturbed samples from expert ones—penalizing them appropriately—while not being so narrow as to overly suppress reward values for slightly perturbed but still useful samples.

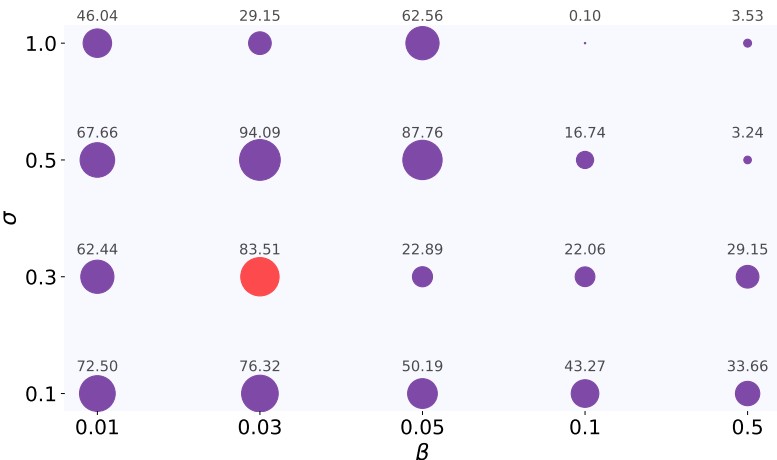

Figure 11: Normalized return for SparseQL+SR-Reward for different hyperparameters $\beta$ and $\sigma$ of the negative sampling strategy. The red circle indicated the return from the hyperparameters used for our experiments in Table 1

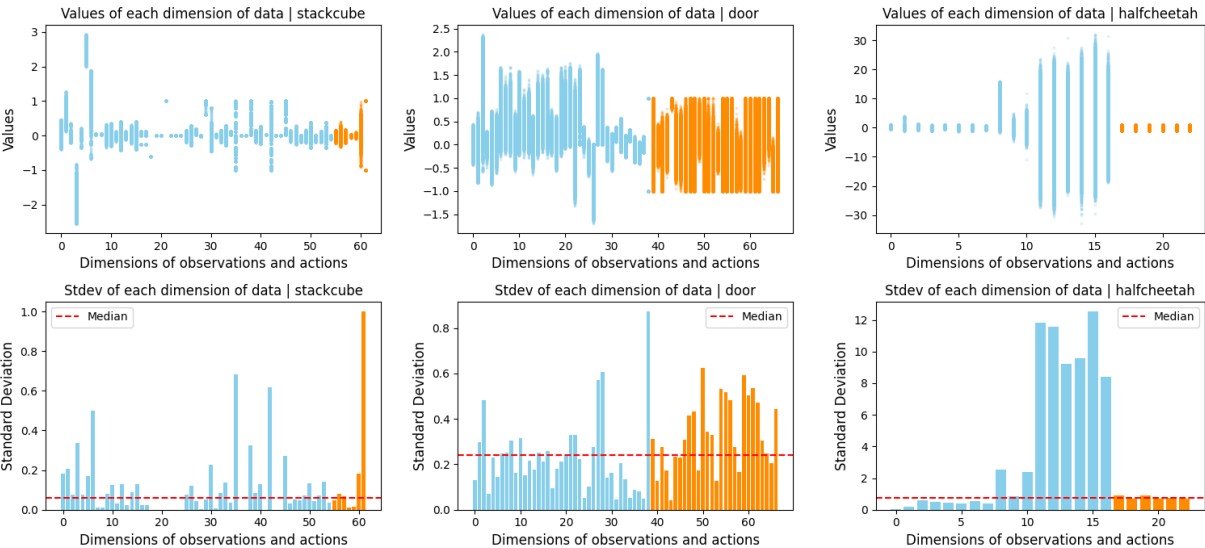

Figure 12: (TOP): Values of every dimension of observation (blue) and actions (orange) for 50 demonstrations of StackCube, Door, and HalfCheetah environments. (BOTTOM): Standard deviation of each observation and action dimension for 50 demonstrations for the same environments. The red line indicates the median of the standard deviations along all dimensions of observations and actions.

# H    Online Reinforcement Learning

SR-Reward can be trained independently of the reinforcement learning (RL) algorithm using only the offline dataset. In this section, we address the question of whether SR-Reward, once trained offline, can be effectively used as a reward function for online reinforcement learning. Figure 13 compares the performance of TD3 trained using the environment's true reward versus using SR-Reward trained on an offline dataset of demonstrations. The TD3 agent was trained on the HalfCheetah environment for 1 million gradient steps using online interactions.

The results highlight a key limitation of SR-Reward in online settings. While incorporating negative sampling improves its robustness to some extent, there is still a notable drop in performance when SR-Reward is used in place of the environment's true reward. This outcome is expected: since SR-Reward is trained solely on expert demonstrations, it does not generalize well to parts of the state-action space that lie far from the distribution of the offline dataset. As the online agent explores new regions during training, SR-Reward may assign incorrect values to unfamiliar state-action pairs, potentially misleading the agent and resulting in suboptimal behavior.

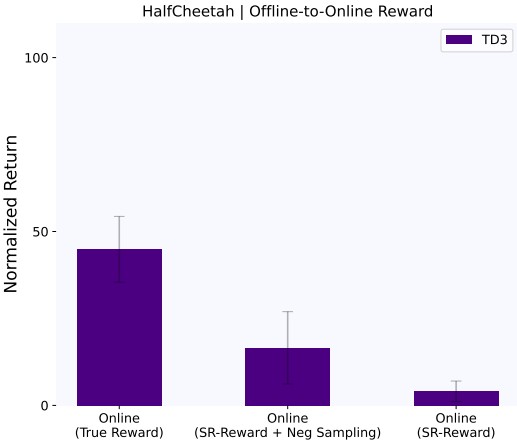

Figure 13: Normalized reward of TD3 algorithm trained using true reward and SR-Reward.

