# OpenReview forum: "SR-Reward: Taking The Path More Traveled"
_TMLR — Accepted by TMLR_

### Review · Reviewer_fZcR · 2025-03-07

**Summary Of Contributions:**

The paper presents a new method, SR Reward, to infer a reward function from demonstration data containing states and actions. The reward function is learned via a Successor (Features) Representation (SR) and is computed as the l2-norm of the successor feature representation. When learning the SR network, an auxiliary objective is used to make sure the state-action representations is predictive of the next states. Negative sampling is used to prevent the reward model from assigning too much value to unseen state action pairs. This takes the form of an auxiliary loss that reaches the reward model to decay the value assigned to a state-action pair as a function of how dissimilar it is to the state-action pairs in the expert demonstrations. The approach is distinct from inverse reinforcement learning because the policy and reward learning is decoupled.

The results demonstrate that using offline RL with the SR Reward performs better than behavior cloning. The benefits of the negative sampling auxiliary loss is verified qualitatively through visualizations of the reward with and without negative sampling in a 2D toy maze environment and by ablating the negative sampling loss on three robotic manipulation tasks.

**Audience:**

Yes

**Broader Impact Concerns:**

I do not see any to warrant concerns about the ethical implications of the work.

**Claims And Evidence:**

Yes

**Requested Changes:**

1. Please fix the formatting issues on pages 7 and 9 where the text stops partway across the page and then there are large blank spaces.
2. To accompany the claims that, "[t]hese findings underscore the versatility of SR-Reward as a modular component that ban be integrated with any offline RL algorithm," make sure to discuss both the offline RL algorithms in the sections above. Right now only sparseQL is discussed. This is important to discuss as there are different performance gaps between BC and f-DVL versus BC and sparseQL. Additionally, the claim that SR-Reward can be integrated with "any offline RL algorithm" should be softened as only two are evaluated.
3. It should be stated how the states were represented for training the SR-Reward and for training the policies. Where image renderings used? Or were joint positions used?
4. Introduce the experiments referenced in Figure 9 prior to the conclusion.
5. Reference the data set size ablations in the main body of the paper and provide some take aways. This is a key analysis.
6. Add a citation supporting the claim, "In contrast, demonstrating the desired behavior is generally more straightforward."

**Strengths And Weaknesses:**

**Strengths**:
1. The method is fairly straightforward, but appears to improve offline RL performance compared to BC for some offline RL algorithms.
2. SR-Reward is evaluated in the context of two different offline RL algorithms.
3. A variety of tasks and task difficulties is assessed.
4. The impact of negative sampling and the corresponding design choices are ablated.
**Weaknesses**:
1. Table 1. is a bit difficult to parse and pull out the key pieces to the story. This makes it difficult to assess exactly how much SR-Reward improves performance. It might better to report the difference in returns between BC vs. SR-Reward and True Reward vs. SR-Reward. The analysis talks about the performance gap between the SR-Reward and the other two methods, so showing the gap will make that easier to follow.
2. There is not inverse reinforcement learning background nor is an explanation given for why one is not provided.
3. There is no mention of the dataset size experiments in the main body of the paper. Given that in contrasting BC versus SR-Reward it is called out that BC performs well when there is a lot of data, it is important to talk about the impact of training dataset size between the two methods. Is the amount of data trained on of the size that BC is expected to perform well? Despite the claims that it is easier to provide a demonstration than to write a reward function, providing many, expert-level demonstrations can be a non-trivial task. So then how much does SR-Reward help to address this. At a minimum, reference Figure 8.
4. It is not discussed how sensitive SR-Reward is to its hyper-parameters nor is the sensitivity of f-DVL and sparceQL to their hyper-parameters when training on the True versus SR-Reward.
5. It is not discussed how state-space complexity and diversity impact the quality of the learned SR-Reward. The tasks evaluated on are rather simple in terms of the diversity of the states, e.g. there the background is fixed across episodes.
6. Figure 1 should detail the main contributions of the paper: (1) the successor representation and (2) the negative sampling. The current figure only showing that the data is augmented is unnecessary.
7. It appears that the SR-Reward method does not always beat BC when the True Reward does, especially for f-DVL.

---

> ### Author Response · Authors · 2025-04-24
> **Response to Requested Changes**
>
> **1) Formatting issues of pages 7 and 9:**
>
> We apologize for the formatting issues in the previous version of the manuscript. These issues on pages 7 and 9 have now been corrected. We have also reviewed the entire manuscript to ensure consistent formatting throughout.
>
> **2) Further clarification about the modularity of SR-Reward and its combination with any RL algorithm:**
>
> We have included the previously missing reference to the results of f-DVL and clarified that our empirical results are limited to two offline-to-online RL algorithms. Accordingly, we have softened our claim regarding the generality of SR-Reward.
>
> **3) State Representation (states or images):**
>
> Thank you for your comment. We have clarified in Section 4.1 (Experiments → Setup) that we use state representations rather than raw images.
>
> **4 & 5) Reference to ablation studies in the main text:**
>
> We have added a paragraph in Section 4.2 that refers to the ablation studies on data size and data quality. The paragraph also briefly discusses the insights drawn from these experiments.
>
> **6) Citations for the claim regarding the collection of demonstrations:**
>
> We have added citations in Section 2.1 (Introduction) to support the claim that providing demonstrations is often easier than specifying reward functions. These references draw from work on collecting demonstrations for real-world robots in the context of imitation learning.
>
> ---
>
> **Additionally:**
>
> **Hyperparameter sensitivity for Negative Sampling:**
>
> To address concerns regarding the sensitivity of negative sampling hyperparameters, we have conducted additional experiments included in Appendix G. These experiments evaluate SR-Reward’s performance on the StackCube task under varying negative sampling hyperparameters, and we offer a rule of thumb for selecting these parameters based on our findings.
>
> **Usefulness of Figure 1**:
>
> While Figure 1 is intentionally simple, we believe it plays a valuable role in helping readers grasp the concept of SR-Reward. It provides a high-level overview of how SR-Reward is used within the offline RL framework. For better integration with the text, we have slightly reduced the figure’s size to make it less intrusive while retaining its illustrative purpose.
>
> ---
>
> We sincerely thank the reviewer for their thoughtful and constructive feedback. We believe that the revisions made in response have significantly improved the clarity and quality of the manuscript.

---

### Review · Reviewer_aD43 · 2025-03-10

**Summary Of Contributions:**

This paper provides an alternative approach (different from the standard inverse reinforcement learning (IRL)) for learning the reward function in offline RL settings. The key insight for the proposed SR-Reward method is to use successor representations as a proxy for feature (of state-action pairs) frequencies pertaining to the demonstration policy. To address the problem of overestimation in out-of-distribution experiences, the paper proposes reducing the rewards obtained to induce a conservative bias as done previously. The effectiveness of SR-Rewards are demonstrated by using them for standard offline RL training (with comparisons to training using ground truth rewards) in classic offline testbeds. The paper also provides comparisons of learning with SR-Rewards to imitation learning (behaviour cloning) in classic offline testbeds.

**Audience:**

Yes

**Claims And Evidence:**

Yes

**Requested Changes:**

Please highlight the changes in response to the previous reviews. Also please address the other weaknesses listed in my review.

**Strengths And Weaknesses:**

Strengths:

The paper addresses an interesting problem of learning rewards using successor representations in offline RL. The proposed technique is indeed intuitive and novel.

Weaknesses:

Since this paper is a resubmission, I went through the entire paper to check if the problems highlighted by the previous set of reviews have been adequately addressed. Unfortunately, I think this is not the case. Particularly, the previous action editor has highlighted three important requirements:

1. Testing the method on tasks where the advantages can be appreciated.

 2. Theoretical justification for why this method works.

3. Considering the hybrid offline-online setting.

This resubmission should have highlighted the changes in response to these comments. However, these details are missing. I would like the authors to discuss the changes in their paper in response to these comments.

In addition, I have the following points to highlight in the paper.

1. There are several design choices in Section 3 which do not have adequate experimental support to showcase its effectiveness. I expect to see more ablation studies to better understand the advantages of the design choices. For example, it is mentioned that an extra reward was added to penalize rewards greater than 1, but no experimental justification is included in the paper. Another example is the use of auxiliary prediction tasks, which is lacking details and justification in the paper.

2. I am not convinced that the experiments are conclusively capturing the intended outcomes. The authors aim to show that SR-Rewards can replace the true reward signal. To demonstrate this they compare the performances of two offline RL algorithms using the true reward and SR-Rewards. However, this comparison should have used the final policies learned by the offline RL algorithms instead of reward values obtained as done in the paper. If the converged optimal policies are similar using both reward functions, it can be claimed that the SR-Rewards can replace the true rewards as they do not change the optimal policies. The paper instead shows that the reward values are similar, and I am not sure what can be inferred from this observation.

3. If I understood correctly, the paper mentions that the best checkpoint during training is then used for evaluation in Section 4.1. This seems very strange. Why was the best checkpoint used instead of reporting the statistics for all trials?

4. More discussions and experiments are needed to demonstrate the need for a negative sampling strategy. When is this helpful and why? The current experimentations and discussions are very limited and I am not sure if this is indeed helpful in general.

---

> ### Author Response · Authors · 2025-04-24
> **Response to Requested Changes (Part 1)**
>
> **1) What has changed since the last submission?**
>
> The main change in this version of the manuscript is the updated experimental results. We discovered that the high variance in the results reported in the previous version was due to a bug in our plotting code. This issue had understandably weakened the strength of our claims. After fixing the bug and re-running the experiments, we now observe more consistent results that better support our conclusions.
>
> We believe this revision aligns with the reviewer’s suggestion to *test the method on tasks where the advantages can be appreciated*. While the tasks themselves remain unchanged, the improved reliability and consistency of the results now make the benefits of our approach more evident in particularly for tasks of growing complexity.
>
> In addition, we have conducted a limited experiment in the *hybrid offline-online setting*, which has been added to the manuscript and discussed in Appendix H. The results align with our intuition that the current method is better suited to purely offline settings. Since SR-Reward is trained on expert offline data, which only covers a small subset of the state-action space, it fails to generalize well to states encountered during online training. This limitation is now discussed in the main text.
>
> **2) Section 3 - The penalty on the reward magnitude:**
>
> Thank you for raising this point. Our decision to cap the reward at 1 was inspired by prior work, such as the original DQN paper (Mnih et al., 2013), which states in Section 5:
>
> >"Since the scale of scores varies greatly from game to game, we fixed all positive rewards to be 1 and all negative rewards to be −1, leaving 0 rewards unchanged. Clipping the rewards in this manner limits the scale of the error derivatives and makes it easier to use the same learning rate across multiple games."
>
> More recent work in inverse reinforcement learning, such as IQLearn (Garg et al., 2021), also explores various methods for bounding the estimated reward signals, including L2 regularization.
>
> We appreciate the reviewer’s observation and have now added a discussion on this design choice in the manuscript, along with references to relevant prior work.
>
> **3) Section 3 - Auxiliary prediction task:**
>
> The use of auxiliary tasks is motivated by a rich body of work in representation learning, which shows that such tasks can enhance the main task's performance. For example, Ni et al. (2024) propose a self-prediction task in reinforcement learning, where the agent predicts the next latent state in conjunction with minimizing the Bellman error—leading to improved latent representations.
>
> Similarly, Fujimoto et al. (2023) incorporate an auxiliary prediction task to train encoders more effectively, where the latent representations are used to predict the next state embedding.
>
> We thank the reviewer for pointing out the lack of explanation in the original submission. We have now added a discussion of this aspect to the text, along with references to the supporting literature.

---

> ### Author Response · Authors · 2025-04-24
> **Response to Requested Changes (Part 2)**
>
> **4) Are experiments conclusively capturing the intended outcomes?:**
>
> We appreciate the reviewer’s thoughtful observation. They correctly noted that our offline RL algorithms were trained once with the true reward and once with SR-Reward, and their performance was compared.
>
> The reviewer commented:
> >"However, this comparison should have used the final policies learned by the offline RL algorithms instead of reward values obtained as done in the paper. If the converged optimal policies are similar using both reward functions, it can be claimed that the SR-Rewards can replace the true rewards as they do not change the optimal policies. The paper instead shows that the reward values are similar, and I am not sure what can be inferred from this observation."
>
> We believe that clarifying our evaluation criteria will help address this concern. Our goal is to demonstrate that SR-Reward can serve as a viable substitute for the true reward. To this end, we measure the performance of the final policies trained with SR-Reward versus the true reward, using the *average return* over many evaluation rollouts. This metric reflects the quality of the learned policy, not just the instantaneous reward.
>
> This approach is particularly relevant in sparse-reward environments, where a simple success/failure signal often serves as the best available measure of task completion. That said, we agree that it is also valuable to assess whether the resulting policies are behaviorally similar.
>
> To address this, we conducted a new experiment comparing the action distributions of policies trained with true rewards versus SR-Reward on the door-opening task. While both policies achieve similar average returns, we additionally visualize the actions chosen by each policy across 50 demonstration trajectories. The distributions across all action dimensions are nearly indistinguishable, indicating that the policies are behaviorally similar.
>
> This new experiment is presented in Appendix F, along with a discussion in the text. We hope this addresses the reviewer’s concern and clarifies the intent of our evaluations.
>
> **5) Checkpoint Selection:**
>
> Thank you for this important point. We apologize for not providing enough detail on our checkpoint selection protocol. For each training run (across five random seeds), we save the checkpoint that achieves the highest mean return over 25 evaluation rollouts, evaluated every 10,000 steps. This is a common practice in offline RL to mitigate overfitting and instability in the final stages of training.
>
> After selecting the best checkpoint per seed, we evaluate each one on 50 fresh rollouts and report the mean and standard deviation across the five seeds (i.e., one value per seed). This process ensures that each reported value corresponds to the final performance of a model selected using a consistent, seed-local criterion and avoids cherry-picking across seeds or runs.
>
> We have revised Section 4.1 to clarify this process and provide more details on the evaluation procedure. We appreciate the reviewer’s suggestion to improve the clarity of our methodology.
>
> **6) Further discussions on negative sampling:**
>
> We have significantly expanded our experiments on negative sampling and added a new appendix (Appendix G) to explore its effects. These experiments vary the negative sampling hyperparameters (noise level and penalty standard deviation) and evaluate SR-Reward’s performance on the StackCube task.
>
> In addition, we visualize the value distributions across each observation and action dimension for datasets from MuJoCo, ManiSkill, and AdroitHand, showing how these dimensions vary in practice. Based on these insights, we provide recommendations for choosing hyperparameters per dataset.
>
> We also explore the role of negative sampling in the hybrid offline-online setting (Appendix H). Although SR-Reward performs poorly in online learning due to limited coverage in the training data, we observe that negative sampling still improves robustness in this setting.
>
> These additions are now discussed in the text (Appendix H), and we hope they provide a more comprehensive understanding of negative sampling’s impact on SR-Reward’s performance.
>
> ---
>
> We would like to thank the reviewer for their thoughtful and constructive feedback. We believe that the changes made in response have significantly strengthened the manuscript and improved its presentation. We hope the reviewer finds our revisions satisfactory and that they are now able to recommend the paper for publication.

---

> > ### Comment · Reviewer_aD43 · 2025-04-27
> > **Reply to responses**
> >
> > I thank the authors for the detailed responses to my review. The major concern that I had was the lack of details in important areas of the paper, as I had stated in my review. These concerns have indeed been satisfactorily addressed by the authors. I am happy to recommend acceptance.

---

> ### Author Response · Authors · 2025-04-24
> **References**
>
> - Scott Fujimoto, Wei-Di Chang, Edward Smith, Shixiang Shane Gu, Doina Precup, and David Meger. For
> sale: State-action representation learning for deep reinforcement learning. Advances in Neural Information
> Processing Systems, 36, 2024.
>
> - Tianwei Ni, Benjamin Eysenbach, Erfan Seyedsalehi, Michel Ma, Clement Gehring, Aditya Mahajan, and
> Pierre-Luc Bacon. Bridging state and history representations: Understanding self-predictive rl, 2024.
>
> - Volodymyr Mnih, Koray Kavukcuoglu, David Silver, Andrei A. Rusu, Joel Veness, Marc G. Bellemare, Alex
> Graves, Martin Riedmiller, Andreas K. Fidjeland, Georg Ostrovski, Stig Petersen, Charles Beattie, Amir
> Sadik, Ioannis Antonoglou, Helen King, Dharshan Kumaran, Daan Wierstra, Shane Legg, and Demis
> Hassabis. Human-level control through deep reinforcement learning. Nature, 518(7540):529–533, February 2015.
>
> - Divyansh Garg, Shuvam Chakraborty, Chris Cundy, Jiaming Song, and Stefano Ermon. Iq-learn: Inverse
> soft-q learning for imitation. In Marc’Aurelio Ranzato, Alina Beygelzimer, Yann N. Dauphin, Percy Liang,
> and Jennifer Wortman Vaughan (eds.), Advances in Neural Information Processing Systems 34: Annual
> Conference on Neural Information Processing Systems 2021, NeurIPS 2021, 2021, virtual, pp. 4028–4039,
> 2021.

---

### Review · Reviewer_RrXb · 2025-04-12

**Summary Of Contributions:**

The authors introduce SR-Reward, a novel method for learning a reward function directly from offline expert demonstrations, designed explicitly for settings where the true reward is unknown and environment interaction is not possible. The core objective is to improve learning efficiency in Inverse RL. In particular, key ideas presented in the paper include:

1.  **Reward from Successor Representations:** The core idea is to leverage the Successor Representation (SR), specifically Successor Features (SF) in the continuous domain, to derive a reward signal. The paper proposes using the $L_2$ norm of the learned state-action SR vector, $M(s, a)$, as the reward function:    $r_{\theta}(s, a) = || M_{\theta}(s, a) ||\_2 $
    The intuition is that state-action pairs frequently leading to future states visited by the expert (as captured by the SR learned from expert data $D_E$) should receive higher rewards. $M_{\theta}(s, a)$ represents the expected discounted sum of future state-action features $\phi(s, a)$ under the expert's policy.

2.  **Offline Temporal Difference Learning:** SR-Reward is learned using a Temporal Difference (TD) approach based on the Bellman equation for SR/SF:
    $M_{\theta}(s_t, a_t) \approx \phi(s_t, a_t) + \gamma M_{\bar{\theta}}(s_{t+1}, a_{t+1})$
    This allows the SR network (and thus the reward function) to be trained solely from the offline demonstration dataset $\mathcal{D}\_E = \{(s, a, s', a')\}$, concurrently with standard offline RL algorithms, without requiring modifications to the RL agent's core learning process beyond using the generated reward $r_{\theta}(s, a)$. The features $\phi(s, a)$ are typically derived by concatenating learned state features $\phi(s)$ with the action $a$.

3.  **Decoupled Reward and Policy Learning:** Unlike many Inverse Reinforcement Learning (IRL) methods, particularly adversarial ones (like GAIL), SR-Reward learning is decoupled from the learning of the agent's policy $\pi$. This avoids the instability often associated with adversarial training frameworks.

4.  **Negative Sampling for Robustness:** To address the potential for function approximation errors leading to reward overestimation for out-of-distribution (OOD) state-action pairs (which are common in offline settings), the paper introduces a negative sampling strategy. Negative samples $(\tilde{s}, \tilde{a})$ are generated by perturbing demonstration samples $(s, a)$ with noise. A loss term is added to penalize the reward $r_{\theta}(\tilde{s}, \tilde{a})$ for these OOD samples, encouraging it to be lower than the reward $r_{\theta}(s, a)$ of the original sample, scaled by a decay factor $\alpha_{\text{decay}}$ based on their distance in the feature-action space:
    $\mathcal{L}\_{\text{Neg.Sample}} = \mathbb{E}\_{(s,a) \sim \mathcal{D}\_E} [ ( r\_{\theta}(\tilde{s}, \tilde{a}) - \alpha\_{\text{decay}} \times r\_{\theta}(s, a) )^2 ]$
    with $\alpha_{\text{decay}} = \exp\left(-\frac{||\phi(s, a) - \phi(\tilde{s}, \tilde{a})||_2^2}{\sigma^2}\right)$. This introduces a bias to preserve the learned reward function near the demonstrated trajectories.

5.  **Empirical Validation:** The effectiveness of SR-Reward is demonstrated through experiments on benchmark offline RL datasets (D4RL MuJoCo, Adroit) and challenging manipulation tasks without predefined rewards (ManiSkill2). The results show that offline RL agents using SR-Reward achieve performance comparable to agents trained with the true environment reward (when available) and significantly outperform Behavioral Cloning (BC) on complex tasks, highlighting the benefit of combining the learned reward with RL's planning capabilities.

**Audience:**

Yes

**Broader Impact Concerns:**

No significant broader impact concerns.

**Claims And Evidence:**

Yes

**Requested Changes:**

*Please Note: All proposed adjustments are in the spirit of strengthening the work, but none are deemed strictly critical for acceptance given the current level of contribution and validation.)*

1.  **Strengthen Discussion on OOD Limitations:**
    *   *Suggestion:* Explicitly state in Section 7 (Limitations and Future Work) that the learned reward $r_{\theta}(s, a)$ may suffer from unreliable extrapolation far from the support of the expert data distribution $\mathcal{D}_E$, as the negative sampling primarily enforces local conservatism. Useful addition would be other potential choices to elucidate why the given choice of metric is sufficient. Also clarify potential implications for online RL or highly exploratory offline policies.
    *   *Motivation:* Improves transparency about the method's scope and manages reader expectations regarding its applicability beyond the offline setting tested.

2.  **Acknowledge Demonstration Quality Assumption:**
    *   *Suggestion:* Add discussion acknowledging that SR-Reward, by rewarding proximity to the demonstrators visitation $||M_{\theta}(s,a)||_2$, implicitly assumes $\mathcal{D}_E$ predominantly contains desirable or near-optimal behavior with sufficient coverage. Noting that performance could degrade if demonstrations are mixed with exploratory/potentially suboptimal trajectories.
    *   *Motivation:* Clearly states a key underlying assumption influencing the quality of the learned reward signal.

3.  **Further Discussion on Theoretical Basis of $L_2$ Norm Reward:**
    *   *Suggestion:* Either adding more formal connections of a brief discussion (perhaps in Limitations or Conclusion) acknowledging that the choice of the $L_2$ norm as the reward function, while empirically successful, currently lacks a formal theoretical justification linking its maximization directly to the recovery of the optimal policy for the unknown ground-truth task reward, unlike prior works in the IRL paradigms (e.g., MaxEnt).
    *   *Motivation:* Solidifying core design decisions  and highlights an area for future investigation.

4.  **Include Hyperparameter Sensitivity Analysis (Appendix):**
    *   *Suggestion:* Provide a brief sensitivity analysis in the Appendix for the key hyperparameters introduced, particularly the negative sampling parameters $\beta$ (noise std dev) and $\sigma$ (decay kernel width), showing performance variation on one or two representative environments as these parameters change around their reported optimal values.
    *   *Motivation:* Increases confidence in the robustness of the empirical results and provides practical guidance regarding HP tuning stability.

5.  **Expand Related Work Comparison to Non-Adversarial Offline IRL:**
    *   *Suggestion:* Enhance Section 5 (Related Work)  and potentially relevant baseline, by briefly discussing and differentiating SR-Reward from other relevant *non-adversarial* techniques for offline reward inference or IRL (e.g., methods based on preferences, contrastive learning, or direct value function inversion like offline adaptations of IQ-Learn).
    *   *Motivation:* Better contextualizes SR-Reward within the broader landscape of modern offline reward learning approaches beyond BC and adversarial methods.

**Strengths And Weaknesses:**

**Strengths:**

1.  **Reward from Successor Feature Norm:** The paper proposes a novel reward function derived from Successor Features (SF), $M_{\theta}(s, a) \in \mathbb{R}^k$, learned from offline expert data $\mathcal{D}\_E$. Specifically, $r_{\theta}(s, a) = ||M_{\theta}(s, a)||\_2$, where $M_{\theta}(s, a)$ approximates the expected discounted sum of future feature vectors $\phi(s', a')$ under the expert policy $\pi_E$. This differs from standard SF usage which typically employs learned linear weights $w$ to recover task-specific value functions $\Big( Q(s, a) \approx M(s, a)^\top w \Big)$. Using the $L_2$ norm provides a task-agnostic measure of expected future feature occupancy density under $\pi_E$.
2.  **Offline TD-based SR Learning:** The SF network $M_{\theta}$ is trained via minimizing the Temporal Difference (TD) error based on the Bellman equation for SFs, using transitions $(s, a, s', a')$ sampled from $\mathcal{D}\_E$:
    $\mathcal{L}\_{\text{Bellman}} = \mathbb{E}\_{(s,a,s',a') \sim \mathcal{D}\_E} \Big[ || M_{\theta}(s, a) - (\phi(s, a) + \gamma M_{\bar{\theta}}(s', a')) ||^2 \Big]$
    This TD-based learning integrates naturally with offline RL algorithms that also rely on TD updates (e.g., IQL, f-DVL, SparseQL), enabling concurrent training without fundamentally altering the RL algorithm's optimization process, beyond substituting the reward signal.
3.  **Stable Non-Adversarial Learning:** The reward $r_{\theta}(s, a)$ is learned independently of the agent's policy $\pi_{\text{RL}}$, avoiding the complexities and potential instabilities of adversarial frameworks (e.g., GAN-like optimization in GAIL involving $\min_{\pi} \max_{D}$ objectives). The optimization involves standard supervised/TD loss minimization for $M_{\theta}$ and standard RL objectives for $\pi_{\text{RL}}$ using $r_{\theta}$.
4.  **Feature-Space Negative Sampling for OOD Regularization:** The proposed negative sampling strategy combats reward overestimation near the data manifold. By generating perturbed samples $(\tilde{s}, \tilde{a}) \sim \mathcal{N}((s,a), \beta^2 I)$ and minimizing $\mathcal{L}\_{\text{Neg.Sample}} = \mathbb{E} \Big[ ( r_{\theta}(\tilde{s}, \tilde{a}) - \alpha_{\text{decay}} \times r_{\theta}(s, a) )^2 \Big]$ with $\alpha_{\text{decay}} = \exp\left(-\frac{||\phi(s, a) - \phi(\tilde{s}, \tilde{a})||_2^2}{\sigma^2}\right)$, it explicitly encourages lower reward estimates for samples diverging from the demonstrations *in the learned feature space $\phi$*. This provides targeted regularization against function approximation errors for OOD inputs close to the support of $\mathcal{D}_E$.
5.  **Empirical Results on Diverse Tasks:** The method demonstrates efficacy by integrating with state-of-the-art offline RL agents (f-DVL, SparseQL) and achieving high performance on D4RL benchmarks (matching or exceeding true reward performance, especially on Adroit) and challenging ManiSkill2 tasks (significantly outperforming BC), validating its utility in practical reward-free offline scenarios.
6.  **Well-Defined Architecture and Training:** The SR network architecture (Fig 2: Encoder, MLP, auxiliary Predictor) and the composite loss function ($\mathcal{L}_{\text{Total}}$ including Bellman, prediction, magnitude penalty, and negative sampling terms) are clearly specified (Sec 3.1, 3.4, Alg 1), providing a concrete implementation blueprint.

**Potential Areas of Improvement:**

1.  **Limited Scope of OOD Constraint:** The negative sampling mechanism primarily imposes conservatism locally around the expert trajectories. For state-action pairs $(s,a)$ far outside the support of the expert distribution $\rho^{\pi_E}(s, a)$, the reward estimate $r_{\theta}(s, a)$ relies mainly on the extrapolation behavior of the $M_{\theta}$ network driven by $\mathcal{L}_{\text{Bellman}}$, and lacks explicit regularization. This might be insufficient for online settings or policies that explore significantly beyond $\mathcal{D}_E$, potentially leading to large extrapolation errors compared to methods with global OOD penalties (e.g., CQL).
2.  **Sensitivity to Expert Occupancy Measure:** SR-Reward fundamentally reflects the discounted state-action occupancy $\rho^{\pi_E}(s, a)$ of the expert. If the expert demonstrations contain significant portions of suboptimal behavior, these regions will still yield high $||M_{\theta}(s, a)||_2$ and thus high reward, potentially misleading the RL agent. The method lacks an explicit mechanism to filter or down-weight such suboptimal data segments.
3.  **Theoretical Justification for $L_2$ Norm Reward:** While the $L_2$ norm of $M_{\theta}(s,a)$ correlates with expert visitation frequency, its optimality as a surrogate reward lacks rigorous theoretical grounding. It's unclear to me if maximizing $\mathbb{E}\_{\pi_{\text{RL}}}[\sum \gamma^t ||M_{\theta}(s_t, a_t)||\_2]$ directly optimizes for the underlying unknown true task reward, unlike, e.g., MaxEnt IRL frameworks which often connect to matching occupancy measures via $r(s, a) \propto \log \pi_E(a|s)$ or similar forms. Establishing a more formal relationship between the proposed objective and standard MDP objectives would be useful.
4.  **Dependence on Feature Quality $\phi(s)$:** The effectiveness hinges on the quality of the learned state representation $\phi(s)$ from the Encoder. The Bellman updates for $M_{\theta}$ and the distance calculations in $\alpha_{\text{decay}}$ rely critically on these features accurately capturing state similarities relevant to the dynamics and expert behavior. The choice of the auxiliary prediction task is heuristic, and alternative representation learning objectives could potentially yield better performance.
5.  **Comparison to Related Offline Reward Inference:** The paper could benefit from a more detailed discussion and potential empirical comparison with other non-adversarial offline reward inference methods. Examples include techniques based on inferring rewards from learned Q-functions (e.g., variants of IQ-Learn adapted for offline use) or contrastive methods that learn rewards by discriminating expert data from non-expert data (which might be synthesized or come from a different source in the offline setting).

---

> ### Author Response · Authors · 2025-04-24
> **Response to Requested Changes**
>
> **1) Discussion on OOD Limitations:**
>
> Thank you for your comment. To further analyze the limitations of SR-Reward, we have added a new study in Appendix H, where we compare the performance of an online RL algorithm (TD3) using SR-Reward versus the true reward in a hybrid offline-online setting. The results indicate that SR-Reward’s performance is bounded by the quality and coverage of the offline data and does not reach the performance of true reward supervision. However, augmenting SR-Reward with negative sampling still yields performance improvements, even in the online setting. This suggests that negative sampling remains beneficial, although overall performance remains constrained by the limitations of the offline dataset. We have included a discussion of these results in the main text, and we hope this study adequately addresses the reviewer’s concerns regarding the OOD limitations of SR-Reward.
>
> **2) Demonstration Quality Assumption:**
>
> We appreciate the reviewer’s point. A brief discussion of the assumption regarding expert-quality data is included in Section 7 (Limitations and Future Work). Furthermore, the new discussion in Appendix H (Online Reinforcement Learning) expands on this limitation by highlighting how SR-Reward's performance is sensitive to the quality of demonstrations, particularly in online settings. We believe these additions help clarify the scope and limitations of our approach.
>
> **3) Discussion on Theoretical Basis of L2-Norm Reward:**
>
> Our work primarily adopts a practical approach to reward construction. As such, we have not included a formal theoretical analysis of SR-Reward. However, we acknowledge the importance of this perspective and, in response to your suggestion, have now explicitly mentioned in the manuscript that the use of the L2 norm to convert the SR vector to a scalar reward lacks theoretical justification. We leave this as an interesting direction for future research.
>
> **4) Hyperparameter Sensitivity Analysis:**
>
> Thank you for your suggestion. We have conducted a sensitivity analysis presented in Appendix G (Negative Sampling Hyperparameters). In this study, we evaluate SR-Reward’s performance on the StackCube task under different configurations of negative sampling hyperparameters. Specifically, we vary the noise level and penalty standard deviation (i.e., decay kernel width). To better understand the characteristics of the datasets, we also visualize the distribution of values across each observation and action dimension in samples from the MuJoCo, ManiSkill, and AdroitHand datasets. Based on these insights, we propose a rule of thumb for selecting negative sampling hyperparameters. We believe this study addresses your concern and provides practical guidance for future users of SR-Reward.
>
> **5) Related Work Comparison to Non-Adversarial Offline IRL:**
>
> Thank you for the valuable suggestion. We have expanded the Related Work section to include additional non-adversarial offline IRL approaches, particularly those relevant to fine-tuning large language models. We believe this enriches the context of our work and highlights the broader applicability of non-adversarial methods.
>
> ---
>
> We sincerely thank the reviewer for their clear and constructive feedback. It was extremely helpful to see suggestions accompanied by their underlying motivation, which made it clear in which direction to improve the paper. We truly appreciate this thoughtful approach. We hope that the revisions made in response to the reviewer’s comments have significantly improved the clarity and quality of the manuscript.

---

### Author Response · Authors · 2025-06-12
**Acknowledgment and Camera-Ready Submission**

We would like to sincerely thank the reviewers and the Action Editor for their thoughtful feedback, constructive suggestions, and engaging discussions throughout the review process. Their input has greatly improved the clarity and rigor of our work.

We are thrilled that our paper has been accepted, and we appreciate the opportunity to contribute to TMLR. The camera-ready version of the manuscript has now been uploaded.

Thank you again for your time and support.

— The Authors

---

### Decision · Action_Editor_gGQK · 2025-05-23

**Recommendation:** Accept as is

**Comment:**

This papers tackles the IRL problem, and proposes a new way to learn a reward function from offline trajectories using Sucessor Representation.

The reviews were overall positive, highlighting the simplicity and efficiency of the method, and all recommend acceptance.The authors answered to the reviews and updated their paper accordingly. This paper is a re-submission: the main reason it was originally rejected was too much noisy evaluations, which has been addressed in this revision.

I recommend acceptance.

**Audience:**

Yes.

**Claims And Evidence:**

Yes.